# Enabling Clonal Analyses of Yeast in Outer Space by Encapsulation and Desiccation in Hollow Microparticles

**DOI:** 10.3390/life12081168

**Published:** 2022-07-31

**Authors:** Simon Ng, Cayden Williamson, Mark van Zee, Dino Di Carlo, Sergio R. Santa Maria

**Affiliations:** 1Department of Bioengineering, University of California—Los Angeles, Los Angeles, CA 90095, USA; simon.ng@ucla.edu (S.N.); caywilliamson@g.ucla.edu (C.W.); mvanzee2014@ucla.edu (M.v.Z.); 2Space Life Sciences Training Program, NASA Ames Research Center, Mountain View, CA 94035, USA; 3Department of Mechanical and Aerospace Engineering, University of California—Los Angeles, Los Angeles, CA 90095, USA; 4California NanoSystems Institute (CNSI), University of California—Los Angeles, Los Angeles, CA 90095, USA; 5Space Biosciences, NASA Ames Research Center, Mountain View, CA 94035, USA; 6KBR, Fully Integrated Lifecycle Mission Support Services, Mountain View, CA 94035, USA

**Keywords:** single-cell analysis, *Saccharomyces cerevisiae*, space biology, microfluidics, cell encapsulation, microbes, preservation, hydrogel

## Abstract

Studying microbes at the single-cell level in space can accelerate human space exploration both via the development of novel biotechnologies and via the understanding of cellular responses to space stressors and countermeasures. High-throughput technologies for screening natural and engineered cell populations can reveal cellular heterogeneity and identify high-performance cells. Here, we present a method to desiccate and preserve microbes in nanoliter-scale compartments, termed PicoShells, which are microparticles with a hollow inner cavity. In PicoShells, single cells are confined in an inner aqueous core by a porous hydrogel shell, allowing the diffusion of nutrients, wastes, and assay reagents for uninhibited cell growth and flexible assay protocols. Desiccated PicoShells offer analysis capabilities for single-cell derived colonies with a simple, low resource workflow, requiring only the addition of water to rehydrate hundreds of thousands of PicoShells and the single microbes encapsulated inside. Our desiccation method results in the recovery of desiccated microparticle morphology and porosity after a multi-week storage period and rehydration, with particle diameter and porosity metrics changing by less than 18% and 7%, respectively, compared to fresh microparticles. We also recorded the high viability of *Saccharomyces cerevisiae* yeast desiccated and rehydrated inside PicoShells, with only a 14% decrease in viability compared to non-desiccated yeast over 8.5 weeks, although we observed an 85% decrease in initial growth potential over the same duration. We show a proof-of-concept for a growth rate-based analysis of single-cell derived colonies in rehydrated PicoShells, where we identified 11% of the population that grows at an accelerated rate. Desiccated PicoShells thus provide a robust method for cell preservation before and during launch, promising a simple single-cell analysis method for studying heterogeneity in microbial populations in space.

## 1. Introduction

Microorganisms are studied in outer space for numerous reasons, including biotechnology development, model organism studies, and pathogen research. As an example of biotechnology development, the BioNutrients missions on the International Space Station (ISS) aim to use engineered *Saccharomyces cerevisiae* and other microbes for the on-demand production of nutrients to be consumed by astronauts on long, deep-space missions [1,2]; the ISS BioRock experiment studied the ability of three microorganisms (*Sphingomonas desiccabilis*, *Bacillus subtilis*, and *Cupriavidus metallidurans*) to aid the biomining of vanadium for in situ resource utilization [3,4]; on the Moonshot Artemis-1 payload, *Chlamydomonas reinhardtii* algae will be grown during transit around the Moon and screened for high producers of lipids and hydrogen [5]. For model organism research, the BioSentinel Artemis-1 CubeSat will expose wild-type and mutant *S. cerevisiae* to galactic cosmic radiation and assays for metabolism and growth to better understand the biological response to deep-space radiation [6,7]. For pathogenesis research, the EcAMSat CubeSat used a pathogenic strain of *Escherichia coli* to evaluate whether microgravity affects antibiotic resistance in low Earth orbit [8]. In each example, microorganisms are studied as a bulk population, with each sample made up of data obtained from thousands of cells concurrently. Thus, measurements represent the averaged behavior of all cells in a sample, which can mask diverse phenotypic properties of individual cells or subpopulations [9,10]. Furthermore, each example relies on no more than 16 bulk samples per condition or timepoint. Single-cell analysis methods, in contrast, enable high-throughput characterizations of hundreds of thousands to millions of individual cells in the population, allowing the identification of subpopulations and rare individual cells.

There are numerous tools for analyzing and manipulating single cells, each with their own applications and limitations. Two well-established single-cell tools are flow cytometry and fluorescence-activated cell sorting (FACS), which can characterize and sort single cells by cell size and/or presence of a fluorescent marker [11]. However, these tools are unable to characterize time-dependent phenotypes such as growth rates of clonal colonies, the cell cycle, or pathogenesis. Additionally, the analysis of cell secretions with FACS is limited, requiring additional steps such as conjugating antibodies to the cell surface, where there can be cross talk between cells without additional precautionary steps [12,13].

Partitioning single cells into uniform-volume compartments addresses these limitations, essentially isolating each cell in its own test tube. Standard high-throughput microwell plates have 96, 384 or even 1536 wells, but these ultimately have limited throughput when compared to high-throughput microfluidic technologies such as water-in-oil droplet emulsions, which can be used to characterize over 100,000 cells per screen [14]. However, many applications using water-in-oil droplets require microfluidically co-encapsulating multiple reagents, such as hydrogel beads for barcoding or secretion binding, increasing the number of encapsulation steps and/or reducing the percentage of droplets containing all reagents. Additionally, the lack of continuous solution exchange in water-in-oil droplets limits the replenishment of nutrients and the elimination of cellular wastes, limiting the growth potential of encapsulated cells [15,16]. A similar approach uses solid hydrogel microparticles (e.g., gel microdrops), but the polymer mesh can interfere with assays, especially those involving genomic DNA [16]. In addition, gel microdrops can limit the growth of microbes in the microparticle and can be degraded prematurely by proteases and amylases released by the encapsulated microbes [17].

A more advanced method combines droplets and microparticles into hydrogel microparticles where an open cavity in the particle captures an aqueous droplet containing a single cell (e.g., nanovials). The hydrogel provides a surface for binding and detecting secretions, similarly to a hydrogel bead in a droplet, while avoiding the co-encapsulation step required for beads [18,19]. However, keeping bound cells from dislodging during long assay or growth protocols and changes in the droplet solution remains challenging. Furthermore, many microbial experiments in space, including BioNutrients, BioRock, and BioSentinel, desiccate their microorganisms prior to launch to increase long-term stabilities and to control microbe metabolic activation via rehydration [1,4,20]. Encapsulating or loading rehydrated cells into water-in-oil droplets or nanovials in space would be an additional hurdle in translating these single-cell analyses to space biology research.

The alternative proposed here is fully closed hollow-shell hydrogel microparticles, or PicoShells, which can be desiccated with single microbes pre-encapsulated inside. PicoShells have a thin hydrogel shell around a hollow, aqueous core, where single cells can be encapsulated. The fully closed geometry traps the cell and its progeny inside the particle core, eliminating the need to emulsify in oil. Additionally, the hydrogel shell is porous, enabling a continuous exchange of nutrients, wastes, and assay reagents with the external environment without needing to destabilize the particle’s structure. The continuous replenishment of nutrients and the elimination of cellular wastes in the particle core allow uninhibited cell growth, more faithfully replicating suspension culture growth conditions and enabling the study of time-dependent phenotypes over days rather than only fast cellular responses. Meanwhile, the diffusion of assay reagents allows solution changes for multi-step assays (i.e., single-cell PCR or ELISA) or even multiple sequential assays on the same cells [15,16]. The hydrogel shell also provides a surface to conjugate various desired functional groups such as capture antibodies. Finally, PicoShells are compatible with fluorescent-activated cell sorters (FACS) such that PicoShells containing single cells or colonies can be selected based on more complex phenotypes than non-encapsulated cells. Compatibility with liquid handling and flow sorting enables the automation of assay processes to reduce crew hands-on time or to enable use in uncrewed autonomous missions such as CubeSats.

Here, we develop a method to desiccate PicoShells with encapsulated single *S. cerevisiae* cells that maintains particle morphology and yeast health after storage and rehydration. Our work builds on the foundation set by van Zee et al., 2022, which introduced the PicoShell technology. Our method renders hollow-shell microparticle single-cell analyses more feasible for future space biology missions that use desiccation-tolerant microbes. *S. cerevisiae* is an ideal proof-of-concept microbe as it is desiccation-tolerant, a model organism, and a useful species for biotechnology applications. Furthermore, *S. cerevisiae* has been used in prior spaceflight missions such as BioNutrients and BioSentinel. Meanwhile, solid hydrogel microparticles, without voids or encapsulated cells, have been desiccated and rehydrated in oil with surfactant while maintaining a morphology similar to that pre-desiccation [21]. We apply this desiccation strategy to PicoShells with *S. cerevisiae*, desiccating particles in Novec oil with Pico-Surf surfactant under vacuum. We evaluate particle integrity via particle durability, morphology, and porosity of the hydrogel shell, and we evaluate yeast health via viability and growth potential. We further show a proof-of-concept growth-based single-cell assay on rehydrated particles.

## 2. Materials and Methods

### 2.1. Cell Culture

Desiccated wild type (WT) (YBS21-A) and mutant *rad51*Δ (YBS29-1) *Saccharomyces cerevisiae* strains were obtained from the NASA Ames Research Center. These strains were also used on the BioSentinel mission [20]. Both strains are diploid prototrophic derivatives of the W303 background. The mutant *rad51*Δ strain is unable to effectively repair double-stranded breaks in DNA. Both WT and *rad51*Δ strains were cultured in YPD medium (Fisher BioReagents, Fair Lawn, NJ, USA) with 50 µg/mL ampicillin (Sigma-Aldrich, St. Louis, MO, USA) at 30 °C and 300 RPM.

### 2.2. PicoShell Fabrication and Yeast Encapsulation

Hollow-shell hydrogel microparticles (PicoShells) were fabricated using a 4-inlet T-junction microfluidic device, largely following methods from a previous study [15]. T-junction microfluidic devices were fabricated from PDMS (polydimethylsiloxane, Sylgard^TM^ 184, Hayward, CA, USA) using standard soft-lithography techniques [22]. To properly form droplets, the device channels must be hydrophobic, so each device was filled with a 2% solution of trichloro(1H,1H,2H,2H-perfluorooctyl)silane (Sigma-Aldrich, St. Louis, MO, USA) in Novec^TM^ 7500 engineered fluid (Novec oil, 3M^TM^, St. Paul, MN, USA). After 5–10 min, the silane was washed out of the device with Novec oil, followed by vacuum aspiration to dry the device channels. 

All reagents were loaded in separate syringes and pumped via syringe pumps (HA2000I, Harvard Apparatus, Holliston, MA, USA). The outer hydrogel shell is composed of 10 kDa 4-arm PEG-maleimide (10% *w*/*w* in particles, pumped at 2 μL/min, Laysan Bio, Arab, AL, USA) and DTT (1,4-dithiothreitol, 3.08 mg/mL in particles, pumped at 2 μL/min, Sigma-Aldrich, Milwaukee, WI, USA) in PBS pH 6.1. The inner liquid core is composed of 9–11 kDa dextran (20% *w*/*w* in particles, pumped at 4 μL/min, Sigma-Aldrich, Milwaukee, WI, USA) in PBS pH 6.1. The oil sheath comprises Novec oil (pumped at 35 μL/min) with 0.5% Pico-Surf^TM^ (Sphere Fluidics, Cambridge, United Kingdom). PicoShells used in this experiment have ~70 μm outer diameter and ~11 μm shell thickness, compared with a typical *S. cerevisiae* radius of 2–5 μm. 

*S. cerevisiae* were encapsulated at 8 million cells/mL in the dextran phase (average cell concentration per particle, lambda = 0.7). However, we observed that some cells were not retained in the dextran flow during particle fabrication and were found in the oil surrounding the particles, resulting in ~15% of particles containing one or more cells. Of the particles with cells, most contained single cells or 2–3 cells clumped together, which is most likely the progeny of a single cell that started budding just before or during particle fabrication. There were fewer particles containing multiple cells that were not clumped together, which is expected according to Poisson loading, but it is still potentially problematic for single-cell assays since the cells likely represent separate single cells, which could have different phenotypes. After fabrication, particles were left overnight for the hydrogel shell to fully crosslink.

WT and *rad51*Δ biological replicates for viability and growth analyses are comprised of 3 colonies picked from YPD agar plates (3 days on agar) and grown in liquid culture for 2–3 days in the cell culture conditions previously described. Yeast cells were washed 3 times with 1X PBS to remove residual medium and then resuspended in dextran with or without 10% trehalose (Fisher BioReagents, Fair Lawn, NJ, USA) for particle fabrication. Trehalose is a disaccharide and desiccation protectant commonly used to promote long-term yeast viability after desiccation [20].

### 2.3. Pre-Desiccation Preparation

The day after fabrication, particles were washed with 2 mL of Novec oil over a 40 μm filter (Corning^®^, Durham, NC, USA) to remove non-encapsulated yeast leftover from the fabrication process. Particles retained by the filter were recovered by backwashing with 2 mL of fresh Novec oil; then, 6 μL of particles was aliquoted into polypropylene microcentrifuge tubes (Sarstedt, Nümbrecht, Germany) pre-treated with 0.1% Pluronic solution and multiple washes with deionized water. One tube of each replicate was prepared for each timepoint. Next, 10 μL of 0.5% Pico-Surf in Novec oil was added on to the particles to prevent them from beginning to dry before applying the desired desiccation method.

For the non-encapsulated yeast controls, 10 μL aliquots of yeast in a 10% trehalose solution at 10^7^ cells/mL was plated in the bottom edge of wells in 96-well Stripwell^TM^ plates (Costar, Vernon Hills, IL, USA) following the protocol from Santa Maria et al., 2020. The non-encapsulated controls were the same 3 WT replicate yeast colonies used for encapsulation.

Separately, particles from each condition were phase transferred from Novec oil to PBS to serve as non-desiccated controls and to investigate the desiccating process in an aqueous solution. To phase transfer the particles, excess oil from fabrication was removed and Pico-Break^TM^ (Sphere Fluidics, Cambridge, United Kingdom) was added at a 1:1 volume ratio. PBS was added at a 3:1 volume ratio, and the mixture was vortexed and then centrifuged. Pico-Break and oil were aspirated, and the process was repeated, leaving this set of particles in PBS.

### 2.4. Desiccation, Storage, and Rehydration

For vacuum drying, particles in open tubes and non-encapsulated yeast in loosely covered Stripwell plates were placed in a vacuum chamber and connected to a vacuum pump (RVR003H, Dekker, Michigan City, IN, USA) for 72 h. For air drying, tubes and plates were sealed with Breathe-Easy membrane (Sigma-Aldrich, Milwaukee, WI, USA) and then exposed to ~20% relative humidity at room temperature and pressure for 3 days in a Parafilm-sealed box with Drierite desiccant (W. A. Hammond Drierite, Xenia, OH, USA). For freeze drying, samples were placed either in tubes or individual Stripwell wells inserted into tubes and frozen at −80 °C overnight. All tubes were plugged loosely with cotton balls and lyophilized for 24 h (Labconco FreeZone 4.5 L −84 °C Benchtop Freeze Dryer, Fort Scott, KS, USA).

All tubes were closed and well plates were lidded; then, desiccated particles and non-encapsulated yeast were double bagged in Ziploc bags containing Drierite desiccant and stored in a Parafilm-sealed box at room temperature (~20 °C) and ~20% relative humidity according to an Arduino-based temperature and relative humidity sensor (DHT22, Songhe, Shenzhen, China).

Cells and particles were rehydrated with 100 µL of YPD medium unless otherwise noted. Non-encapsulated yeasts were moved directly to a 30 °C incubator for 30 min. Particles were centrifuged briefly to pull the rehydrating solution through the particle clump and then vortexed and vigorously pipetted to break the clump into smaller clumps and individual particles.

### 2.5. Particle Morphology

Particles were imaged at key stages of desiccation: pre-desiccation, post-desiccation, post-storage, and post-rehydration. All microscopy was performed on an EVOS FL Cell Imaging Microscope (AMG^TM^, Mill Creek, WA, USA) except where otherwise noted. For quantitative analysis, particle diameters and shell thicknesses were measured manually in ImageJ (v1.53c) from 40× magnification images. All particle diameters and shell thicknesses were measured along a conserved axis.

### 2.6. Diffusion of FITC-Dextran

To assay diffusion across the PicoShell particles’ hydrogel shell, 20 μL of desiccated particles was rehydrated in 100 μL of PBS pH 7.4 and then mixed 1:1 with 20 kDa, 40 kDa, or 156 kDa fluorescein isothiocyanate-dextran (FITC-dextran, Sigma-Aldrich, Milwaukee, WI, USA) in deionized water to produce a 0.5 mg/mL FITC-dextran solution containing particles. This solution was immediately loaded onto a 100 μm height cellometer (Nexcelom, Lawrence, MA, USA), producing a monolayer of particles, and imaged at 20× magnification with 30 ms GFP-channel exposure at 50–80% illumination. Particles were imaged after incubation with FITC-dextran for 1–2 min (“1 min”), 20 min, 1 h, and 24 h. In ImageJ, an intensity profile cut line was measured across each particle and its immediate surroundings. The intensity within the particle was divided by the intensity outside the particle, producing a relative fluorescence inside the particle. 

### 2.7. Viability and Growth

To assess the viability of non-encapsulated yeasts after a post-rehydration incubation (30 min), YPD medium was carefully removed and 100 μL of stain solution (PBS pH 7.4 with 10 µg/mL fluorescein diacetate (Sigma-Aldrich, Milwaukee, WI, USA) and 5 µg/mL propidium iodide (Invitrogen, Eugene, OR, USA)) was added. After incubation for 10–15 min in the dark at room temperature, an overlay image was taken at 40× magnification with brightfield (100% illumination), GFP, and RFP (60 ms exposure) channels. Live cells (stained green and unstained) and dead cells (stained red) were counted for each image using the Cell Counter plugin in ImageJ. The post-rehydration viability for each yeast culture was calculated as the number of live cells divided by the total number of cells, normalized by the viability of the same culture pre-desiccation. For example, before desiccation, a sample of one of the non-encapsulated yeast colonies had 436 live cells out of 501 total cells (87.0% viability). After desiccation and 3.5 weeks of storage, a rehydrated sample from the same yeast culture had 188 live cells out of 413 total cells (45.5% viability). The normalized viability for the 3.5-week timepoint is 45.5/87.0 = 52.3%. Normalizing helps isolate the effect of desiccation on viabilities for each condition and helps replication by accounting for natural differences in baseline viabilities. 

To assess viability and growth of yeast in particles, rehydrated particles in tubes were diluted in 200 μL of YPD medium, incubated at 30 °C for 16 h, and vortexed every 6 h to ensure access to nutrients. Particles were washed 3 times with PBS, transferred to a 96-well plate (Falcon^®^, Durham, NC, USA), and incubated with 100 μL of stain solution for 20–40 min in the dark at room temperature. Wells were imaged on a Nikon Eclipse TI microscope using a Photometrics camera and NIS-elements AR software. The entire well was imaged with a 6.4 × 6.4 mm^2^ tile image at 10× magnification with brightfield, TRITC, and FITC channels; both fluorescence channels used 200 ms exposure time. The overlaid brightfield, TRITC, and FITC large-scan RGB image was manually annotated in ImageJ using the grid tool and cell counter plugin. Multiple categories of yeast in particles were annotated, including dead yeast (stained red), yeasts that are alive (stained green or unstained) but not growing robustly (fewer than 5–8 cells in the particle), particles with some yeast growth (~8–30 cells), particles full of yeast, and particles swollen with yeast (particle diameter stretched by growing yeast).

Each particle counts as one cell (i.e., a particle containing a colony of thousands of yeast cells only counts as one live cell—the parent of the colony); for viability measurements, all particles with cells are considered “live” unless they are “dead”, and the calculation and normalization is the same as for non-encapsulated cells. For yeast growth potential measurements, particles full of yeast and swollen with yeast are counted as “growing”, and normalization is applied in the same manner as viability. For subpopulation quantification, the population is divided into “swollen” (yeast stretching particle), “grown” (particle full of yeast), “live” (stained green or unstained with a single cell, a few cells, or a small clump of cells), and dead yeast. There is no normalization for subpopulation analyses. 

### 2.8. Clonal Growth

Rehydrated particles were plated sparsely in wells with media and incubated at 30 °C without shaking. Multiple locations were selected within the well to image through time, and the well plate was handled gently to avoid shifting the particles within the well. Selected locations were imaged every 6 h for 24 h, tracking the same individual particles as the yeast within them replicated.

Each particle was tracked through the timelapse series of images, manually accounting for slight movement of particles over time. The number of yeasts in each particle was counted manually until a cell clump formed that spanned multiple focal planes. The number of yeasts in a cell clump was estimated by measuring the approximate area of the clump in ImageJ, treating that area as a circle, calculating the corresponding sphere volume, and then dividing that volume by the approximate volume of an individual yeast cell while finally multiplying it by a spherical packing factor of 0.74. The yeast’s radius was set as 2 μm, providing a yeast volume of 33.5 μm^3^/cell.

### 2.9. Statistics

All statistical tests were performed in GraphPad Prism (v8.3.0). For particle diameters, shell thickness, yeast viability, and yeast growth potential measurements, experimental conditions were compared against themselves within each timepoint with one-way ANOVAs followed by Tukey’s multiple comparisons tests. For particle shell porosity, experimental conditions were compared within and between molecular weights with a one-way ANOVA followed by Tukey’s multiple comparisons tests. Viability and growth potential measurements were also compared against themselves between 0.5 and 8.5 weeks with unpaired T-tests. Subpopulations were compared by unpaired T-tests. Throughout this paper, one asterisk (*) is *p* ≤ 0.05, two (**) represent *p* ≤ 0.01, three (***) represent *p* ≤ 0.001, and four (****) represent *p* ≤ 0.0001.

## 3. Results

### 3.1. Desiccation Procedure

An ideal desiccation procedure preserves PicoShell particle integrity and yeast-cell health over long durations as closely as possible to fresh PicoShells and cells. Of all methods tested here, we found that the vacuum-based desiccation of PicoShells in Novec oil with Pico-Surf caused the least impact on particles and yeast compared to the fresh condition. In this process, particles are fabricated in Novec oil with Pico-Surf according to established protocols [15], with the addition of 10% trehalose in the dextran phase to aid yeast viability. Particles in Novec oil are strained through a 40 μm filter and washed with additional Novec oil, washing away non-encapsulated cells. Particles are recollected in fresh Novec oil, transferred to a microcentrifuge tube or well plate, and a small amount of fresh Novec oil and Pico-Surf is added on top of the particles. The desiccation process is completed by vacuum drying the particles with a compressor pump for 72 h with the tube caps open. To grow and study the yeast, dried particles are rehydrated with liquid YPD medium, which caused nearly instantaneously swelling in the dried hydrogel particles. The desiccation process is shown in Figure 1 and particle rehydration can be observed real time in Appendix A. Rehydration also washes out the dextran and trehalose core of desiccated PicoShells, leaving rehydrated particles with a dextran-free aqueous core (Appendix A) and that are ready to be used for colony growth or other single-cell assays.

Before selecting vacuum drying in oil as our desiccation method of choice, we investigated desiccating PicoShells in aqueous solution rather than oil, but this approach yielded misshapen and highly aggregated particles. PicoShells are fabricated in Novec oil with a Pico-Surf surfactant, but in the typical use of PicoShells, they are phase transferred out of oil into an aqueous solution, similarly to growth media, to grow and study the cells inside. After transferring PicoShells from oil to water, we tried three desiccation methods: vacuum, air, and freeze drying. All three methods yielded deformed, collapsed particles prone to aggregation (Appendix A), which limits the ability to handle or visualize cells or cell growth in individual PicoShells. These results contrasted with particles vacuum dried in oil before phase transferring to aqueous solution, which showed greatly improved morphology and reduced aggregation. 

### 3.2. Post-Desiccation Shelf Life

We rehydrated vacuum-dried particles containing encapsulated yeast at three timepoints over the course of 8.5 weeks to investigate how particle integrity and yeast health change with increasing storage duration. In brief, we observed very little change in the particle’s durability, morphology, and porosity and limited change in yeast viability, although there was a significant decrease in the portion of the population able to grow following rehydration. Additionally, particle aggregation increased with storage duration. Five conditions for two yeast strains were tested and will be compared in subsequent sections: +Trehalose particles: PicoShells dried containing 10% trehalose in the dextran phase and wild type (WT) yeast;−Trehalose particles: PicoShells dried with WT yeast, without trehalose;*rad51*Δ particles: PicoShells dried with 10% trehalose and a mutant yeast strain, *rad51*Δ, deficient for double strand DNA repair;Aqueous particles: PicoShells with WT yeast, never dried, instead phase transferred and kept hydrated in PBS. These particles were stored at room temperature outside the dry box;Non-encapsulated yeast: WT yeast dried free in 10% trehalose, not encapsulated in particles.

#### 3.2.1. Particle Integrity

Compared to non-desiccated aqueous PicoShells, rehydrated PicoShells withstand similar physical manipulations (i.e., vortexing, pipetting, etc.), maintain similar morphology, and have similar porosities of the hydrogel shell. Rehydrated particles that were desiccated in oil do not appear to have tears, defects, or otherwise become damaged during manipulation, representing a vast improvement in utility over particles desiccated after being phase transferred (Appendix A). Rehydrated particles survive vortexing at max speed, vigorous pipetting, centrifugation at 1000 RCF, and flow sorting. Additionally, some yeast colonies grow until they physically stretch the particle, expanding the particle diameter from ~70 μm to over 200 μm in some cases; rehydrated particles can withstand this internal pressure from the yeast pressing on the hydrogel wall, indicating that the hydrogel’s strength is preserved. At larger diameters of ~150 μm, all PicoShells are at risk of rupturing even from gentle manipulations. 

Rehydrated particles retained an intact hollow-shell morphology and were similar in overall shape and circularity to aqueous particles (Figure 2A). After desiccation and storage for 8.5 weeks, rehydrated particles shrunk less than 18% in overall diameter and less than 30% in hydrogel shell thickness compared to non-desiccated aqueous particles (Figure 2B,C). These decreases in diameter and shell thickness for rehydrated particles are statistically significant, but rehydrated samples will not realistically regain the exact same morphology as fresh samples; thus, statistically significant differences are expected and also a less important metric than the magnitude of changes caused by desiccation. Among rehydrated particles, +Trehalose and −Trehalose particles look very similar and only differ significantly for diameter at the 0-week, non-desiccated timepoint (Appendix A), indicating that the presence of trehalose during desiccation has little influence on particle morphology.

Rehydrated and fresh aqueous PicoShells also have similar hydrogel shell porosities. Diffusion across the shell is important for the exchange of nutrients and cellular wastes to support yeast growth and for assay reagents to reach the cells inside particles. These assay reagents could include small molecules, such as stains and drugs, or large molecules, such as enzymes and antibodies. We investigated the shell’s porosity by incubating particles in solutions of fluorescently labeled dextran (FITC-dextran) of varying molecular weights (20 kDa, 40 kDa, or 156 kDa). We measured fluorescence inside and just outside each particle, giving a relative fluorescence that represents diffusion into the particle (Appendix A). The porosity of +Trehalose particles rehydrated 3.5 weeks after desiccation is not significantly different from fresh aqueous particles after 20 min of incubation in 40 kDa and 156 kDa FITC-dextran solutions (*p* = 0.057 and *p* = 0.15, respectively), although the 20 kDa condition does show significant differences (Figure 3). Still, diffusion into rehydrated particles is more similar to diffusion into fresh particles within each size FITC-dextran than between sizes. Therefore, molecules of a given size can be expected to diffuse across the hydrogel shell similarly for rehydrated particles as fresh aqueous particles. In addition to measuring relative fluorescence after 20 min of incubation, we took measurements after 1 min, 1 h, and 24 h of incubation, which show that diffusion profiles over time are similar within each size FITC-dextran (Appendix A). Additionally, −Trehalose has similar diffusion profiles as +Trehalose (Appendix A), indicating that the presence of trehalose during desiccation has little effect on hydrogel shell porosity. In the few ripped particles we observed, an FITC-dextran solution of any size immediately fills the particle (Appendix A). Ripped particles were rare and, thus, not of great concern.

As storage duration increased, rehydrated PicoShells aggregated more, requiring increasingly vigorous pipetting and agitation to break particles apart during rehydration (Appendix A). One week after desiccation, rehydrated particles separate easily from one other and pipetting 20–30 times was sufficient for separating particles. After 3.5 weeks, the clump of particles was large enough to block the orifice of the pipette tip, requiring vigorous pipetting to break up. After 8.5 weeks, the clump had to be sheared into smaller pieces to be pipetted, which was performed with the side of the pipette tip on the microcentrifuge tube wall. Even then, some large particle clumps remained and did not separate with pipetting or vortexing. This treatment does not appear to seriously impact the morphology of individual PicoShells. We did not notice an increase in ripped or deformed individual PicoShells after more vigorous pipetting.

#### 3.2.2. Yeast Health

Yeast desiccated and rehydrated in particles maintained a high viability over 8.5 weeks of storage but showed a decline in the portion of the population able to grow. +Trehalose PicoShells retained the highest viability and growth after 8.5 weeks compared to other conditions, with 85% of the population alive and 15% of the population able to grow and establish colonies (Figure 4). The greater yeast health observed in +Trehalose compared to aqueous and −Trehalose PicoShells indicates that desiccation and trehalose aid yeast health for long term storage. +Trehalose PicoShells show an insignificant decrease in viability over 8.5 weeks (*p* = 0.095) but a significant decrease in growth (*p* < 0.05), although longer-term data are lacking. Interestingly, yeast in +Trehalose particles have more than double the viability of free, non-encapsulated yeast at 8.5 weeks (Figure 4A), indicating that encapsulation does not harm the yeast and that at least one aspect of encapsulation significantly improves yeast viability. The raw, non-normalized data can be observed in Appendix A.

### 3.3. Single-Cell Analysis Potential

We performed proof-of-concept growth-based single-cell analyses on wild type (+Trehalose, WT) and *rad51*Δ yeast populations to investigate heterogeneity in growth characteristics within each population and between the two populations (Figure 5). Both populations were rehydrated and analyzed 0.5 weeks after desiccation (same particles as “0.5 weeks” in Figure 4). We evaluated growth within hundreds of WT and *rad51*Δ particles after 16 h of incubation in YPD medium, demonstrating a simple method to observe the distribution in growth behaviors within a population. WT and *rad51*Δ populations were categorized into four growth-related subpopulations—“swollen” (yeast stretch particle), “grown” (yeast fill particle), “live” (yeast are alive, but not grown or swollen), and “dead”. Over half of the WT population grows to fill or stretch the particle (swollen or grown), around a quarter of the population is alive with little to no growth (live), and only 14% of the population is dead (Figure 5A). Comparing the WT and *rad51*Δ populations, the *rad51*Δ population has significantly fewer growing cells (swollen or grown; *p* < 0.05) and twice as many dead cells. With free yeast not encapsulated in PicoShells, individual colony growth cannot be quantified, and the swollen, grown, and live subpopulations identified here would be commingled and indistinguishable. The ability to identify live but not growing cells is particularly unique, as these cells are obscured by the growing cells in a typical bulk population assay. 

To expand on our subpopulation quantification, we measured clonal growth curves of yeast in 20 WT and *rad51*Δ particles. As seen in Figure 5B, most WT yeast grew to colonies larger than 4000 cells by 24 h, indicating these cells grew robustly and at a similar rate to each other. Other WT yeast within the population did not replicate more than a handful of times (fewer than 10 cells). In contrast, encapsulated *rad51*Δ yeast show larger variation in colony size over time, potentially indicating a spread in growth rate. Four *rad51*Δ clones grew to colonies between 100 and 1000 cells, although at a slower rate than WT yeast. One *rad51*Δ cell grew to a colony of over 7000 cells, equivalent to or larger than WT colonies, and a few other *rad51*Δ cells also grew to large colonies. These rare *rad51*Δ cells could be of interest for further study to understand why they grew so well compared to other *rad51*Δ cells. In a typical bulk population growth rate assay (i.e., optical density), the growth of all cells is effectively summed, so a researcher would assume that all *rad51*Δ cells grew equally mediocrely and would never know that most growth actually came from a few unusual cells. 

## 4. Discussion

### 4.1. Advantages of Desiccated PicoShells

Single-cell analysis offers a high-throughput method to study the heterogeneity within cell populations and to identify rare cells of interest that would otherwise be obscured within a bulk population. Single-cell analysis in space is already gaining traction: 10x Genomics plans to fly their single-cell genomics technology, based on water-in-oil droplets, to the ISS aboard the second Axiom Space ISS mission [23]. The 10x Genomics platform uses water-in-oil droplets with a barcoded hydrogel bead and has single-cell genomic, epigenomic, and transcriptomic applications [14]. PicoShells compartmentalize single cells in permeable hydrogel capsules to provide additional single-cell analysis tools for exploring cell population heterogeneity, offering the ability to track individual cells and clonal growth over the course of days, for stringing multiple assays together in sequence, and for isolating specific live subpopulations for further study. Desiccated PicoShells do not require on-orbit encapsulation equipment and reduce active crew time and workflow complexity compared with water-in-oil droplets or freshly fabricated PicoShells. 

Additionally, changing the solution inside PicoShells is simple. Yeasts are grown upon addition of YPD medium, washed by PBS, and stained for viability, all within the same PicoShell. These abilities improve versatility over other single-cell analysis technologies such as water-in-oil droplets, which cannot easily be exchanged with new reagents or solutions after encapsulation except by techniques such as droplet merging, which require additional microfluidic devices and increased complexity [14].

### 4.2. Desiccation Method

Based on our data, our recommended desiccation method is to vacuum dry particles in Novec oil and 0.5% Pico-Surf surfactant with 10% trehalose added to the particle’s dextran phase during particle fabrication. Desiccating particles in oil before phase transfer to aqueous solution leaves the PicoShell core filled with dextran and trehalose, which are otherwise washed out in aqueous solutions. Desiccated PicoShells are composed of a desiccated yeast cell surrounded by a dry ball of dextran polymer and trehalose disaccharide, all surrounded by the particle’s dried hydrogel shell. During desiccation, the dextran core may provide scaffolding support to the hydrogel shell as water is removed, preventing the hydrogel shell from fully collapsing on itself. This may explain why PicoShells tend to deform more and irreversibly change morphology when desiccated in aqueous solution (Appendix A); the dextran is designed to diffuse out during phase transfer instead of being retained in the inner core similarly to when the PicoShells are never transferred out of oil. Upon rehydration, dextran immediately diffuses across the hydrogel shell into the rehydrating solution and may provide some outward force to help the hydrogel shell re-expand (Appendix A).

Dextran and trehalose likely work together to preserve yeast health. Trehalose forms a vitreous scaffold upon desiccation, which helps prevent cell damage by preserving protein structure and inhibiting free radicals [20]. Meanwhile, dextran has been shown to aid vitrification and improve preservation of proteins both alone and combined with disaccharides by increasing the glass transition temperature [24,25,26]. The combined action of trehalose and dextran during desiccation and storage may explain the improved viability of yeast in +Trehalose particles compared to non-encapsulated yeast (Figure 4A). Desiccation with dextran and trehalose could potentially improve yeast viability for all yeast desiccation applications regardless of encapsulation.

The vacuum desiccation protocol can likely be shortened by desiccating particles for less than 72 h. In some experiments, particles and yeast were vacuum dried for 18 to 24 h, with no apparent difference in dryness under the microscope and during rehydration (Appendix A). However, hydration levels were not rigorously assessed, and long-term particle integrity and yeast viability data would be important to confirm how short the vacuum drying process can be. Besides vacuum drying, air drying and freeze drying (lyophilization) are worth investigating with PicoShells in Novec oil and Pico-Surf before phase transfer to aqueous solution. Previous work with the same WT and *rad51*Δ *S. cerevisiae* strains from this study found that air drying conferred the highest viability after storage for 23–32 weeks, followed by vacuum and then freeze drying [20]. Meanwhile, freeze drying was successfully applied to solid hydrogel particles in Novec oil with Pico-Surf [21].

### 4.3. Potential Future Applications

With further development, PicoShells could be designed to quantify single-cell protein secretions, an important measurement for microbial recombinant protein production [27,28]. Antibodies specific for the secreted proteins of interest can be conjugated to the particle’s hydrogel shell to bind the secretion and prevent diffusion out of the particle, and secretions can be tagged with fluorescent secondary antibodies [18]. The same principle can be applied for other assays—any molecule that can be attached to a reactive thiol or maleimide group can easily be conjugated to the hydrogel shell during fabrication. This can be useful if a user wants to add motifs that capture genetic material or enable cell adherence to the interior of the outer shell. The functionality of conjugated molecules would need to be assessed after desiccation and rehydration, although similar work in which researchers freeze dried hydrogel particles in Novec oil with Pico-Surf showed that methacryloyl and vinylsulfone groups were still reactive after rehydration [21].

Specific cell subpopulations in PicoShells can be isolated using commercial flow sorters, with over 100,000 particles screened per sorting experiment [15]. Flow sorting allows for the isolation of interesting subpopulations based on scattered light and fluorescent readouts [11]. For example, in our clonal growth rate assay in Figure 5B, flow sorting could be used to isolate the fast-growing *rad51*Δ subpopulation from the slow-growing subpopulation, or a population could be sorted based on both growth and production of fluorescently labeled products. Isolated subpopulations remain alive and could be studied further or used as a parent strain for further engineering. Flow sorting could also be used before desiccation to enrich for particles containing a cell; few particles initially contain a cell due to Poisson loading at low loading fractions during particle fabrication. A low loading efficiency ensures that most PicoShells containing cells initially have single cells rather than doublets, triplets, etc. Empty PicoShells can be removed before desiccation via flow sorting, isolating PicoShells containing single cells. This is useful for researchers who want to increase colony sorting throughput post-assay and for flight missions where minimizing experiment mass is critical. In preliminary experiments, rehydrated PicoShells in water have been sorted on a flow sorter, but further experiments would be required to sort PicoShells in oil before desiccation. Flow sorting could be further optimized by using smaller PicoShells. In this study, we use PicoShells that are 70 µm in diameter. However, we can achieve PicoShell diameters between 30 and 50 µm to be more compatible with standard nozzle sizes used in flow sorters [15].

PicoShells can also be engineered to fully degrade and release the cells encapsulated inside. For example, PEG-maleimide PicoShells crosslinked with DTT or matrix metalloproteinase (MMP)-degradable peptides can be degraded by sodium periodate (NaIO_4_) or trypsin, respectively. Meanwhile, PicoShells fabricated from PEG-OPSS (ortho-pyridyldisulfide) and crosslinked with DTT can be degraded by reducing agents such as TCEP (tris(2-carboxyethyl)phosphine) or DTT [15]. While controlled degradation is a useful tool, undesired degradation should be mitigated. Other investigators have shown that fresh PicoShell-like particles made from 8 kDa 4-arm PEG-diacrylate are stable in nearly pure ethanol, methanol, acetonitrile, and acetone and are stable through a standard 35-cycle PCR reaction with a maximum temperature of 98 °C during the denaturation step [16]. Both desired degradation mechanisms as well as chemical and thermal stability should be verified for desiccated PicoShells.

Desiccated PicoShells also have potential for automated and autonomous workflows, which are important for the ISS and prerequisite for any uncrewed missions such as CubeSats [7]. PicoShells could be desiccated and sealed in a chamber of a microfluidic device and later rehydrated via a pump-based delivery system. Particles could be imaged in a monolayer by limiting the height of the microfluidic chamber to just larger than the PicoShell diameter. Preliminary experiments have shown the ability to desiccate PicoShells in a microfluidic device and rehydrate them with a manual syringe. A fully automated workflow with desiccated PicoShells would facilitate single-cell analyses on deep space biological CubeSats and other uncrewed exploration missions to the Moon and Mars.

PicoShells could also benefit the Earth’s fermentation industry by providing a high-throughput assay tool to screen large strain libraries. Strain engineering relies on generating a diverse library of strains and then screening the library in microwell plates or bench-scale fermenters to find strains with high titers, viability, and biomass [29]. Screening hundreds of microwell plates per day is possible for companies with robust laboratory automation but still very labor and resource intensive. One 0.5 mL tube filled with PicoShells can more than replace hundreds of plates. Furthermore, cells in PicoShells could be cultured in a fermenter before high-throughput screening, more closely replicating the eventual production environment than microwell plates and potentially accelerating scale up [15]. If PicoShells became an integral part of the strain engineering workflow, fermentation companies could bring the fabrication process in-house and forgo desiccation, but for academic labs or companies without microfluidics expertise, obtaining desiccated PicoShells from a microfluidics collaborator could be more immediately feasible. 

### 4.4. Limitations

A chief limitation is that the desiccated PicoShell platform is only useful for cells that can survive desiccation. Many microbes can survive at least a short period of desiccation, and some, such as *S. cerevisiae*, can survive for years in a desiccated state, but mammalian cells do not naturally survive desiccation. Ongoing efforts to desiccate mammalian cells via freeze drying are making progress [30], but in the meantime, nanovial microparticles could be more immediately feasible for single-cell mammalian research [18,19]. Unlike desiccated PicoShells, nanovials do not offer cell preservation and have limited ability to study time-dependent phenotypes, but nanovials could still be useful to encapsulate and assay single hydrated mammalian cells in space without on-orbit particle fabrication. Finally, for applications not requiring desiccation, fresh PicoShells can be used for many cell types, including mammalian Chinese hamster ovary (CHO) cells [15].

An experimental limitation in this work is the particle aggregation observed with increasing storage duration (Appendix A). Clumps of PicoShells obscure most particles in the clump during imaging, may limit diffusion to yeast deep within the clump, prevent flow sorting of individual clones, and may contribute to reduced yeast viability and growth after rehydration. While the exact cause of the observed aggregation is unknown, one possibility is that particles contain unreacted maleimide and thiol groups from fabrication, which react to form bonds between particles. During desiccation, the Novec oil surrounding particles evaporates, and particles may come close enough for any unreacted molecules on their surfaces to react. This concept has been suggested before [31], and a possible solution is to block unreacted thiol groups, for example by incubating particles in a solution of N-Ethylmaleimide before desiccation. Particle aggregation may also be reduced by changing the oil type and surfactant concentration or by using alternative mixing methods during rehydration, such as sonication, although an ideal system would only need gentle mixing to rehydrate and de-aggregate the PicoShells. 

Another limitation of our desiccation method is that the growth potential of rehydrated yeast decreases much more than viability (Figure 4), suggesting that while many yeasts in particles remain alive, their metabolism and cell cycle are disrupted. With altered metabolism, rehydrated yeast may behave and perform differently than non-desiccated yeast. However, with proper ground controls, controlling for rehydrated yeast’s altered behavior should be possible.

No desiccation method will perfectly preserve hydrogel microparticles or encapsulated cells. However, the PicoShell desiccation method we present here not only improves post-rehydration viability over non-encapsulated yeast, but it preserves particle integrity and yeast health sufficiently well to enable a single-cell analysis that showed cellular heterogeneity in growth behavior and identified highly performing cells. We foresee desiccated PicoShells facilitating increasingly complex single-cell analyses in outer space microorganism research, including clonal growth rate assays and multi-step assays. Space biotechnologies could use PicoShells to screen hundreds of thousands of cells in high throughput to identify highly performing cells for strain engineering, while model organism and pathogen studies could use this technology to evaluate cellular heterogeneity in the response to space stressors. Desiccated PicoShells work in conjunction with microbial desiccation to preserve cells for long-duration deep space experiments while simultaneously providing a ready-to-use single-cell analysis platform upon rehydration.

## Figures and Tables

**Figure 1 life-12-01168-f001:**
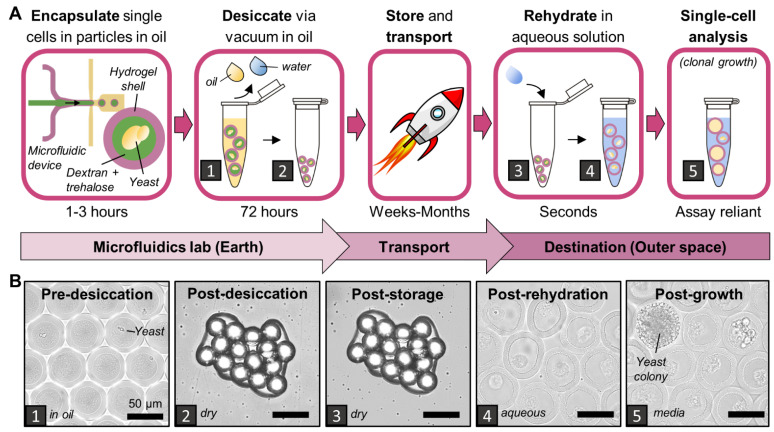
**PicoShell desiccation workflow.** (**A**) Yeasts suspended in a liquid dextran polymer solution are microfluidically encapsulated by a solid hydrogel shell. Particles and encapsulated yeast are desiccated under vacuum to remove oil and water, and then they can be stored and transported. When the time for use comes, particles are rehydrated in water or liquid medium. Particles can be analyzed for colony growth or other single-cell derived colony traits. (**B**) Particles were imaged through each stage of the desiccation process. The number in the bottom left corresponds with the workflow steps in (**A**). The post-desiccation (2) and post-storage (3) images were captured 8.5 weeks apart and are not the same image. The post-rehydration image is of the same particles as in the post-desiccation and post-storage images. All scale bars are 50 μm.

**Figure 2 life-12-01168-f002:**
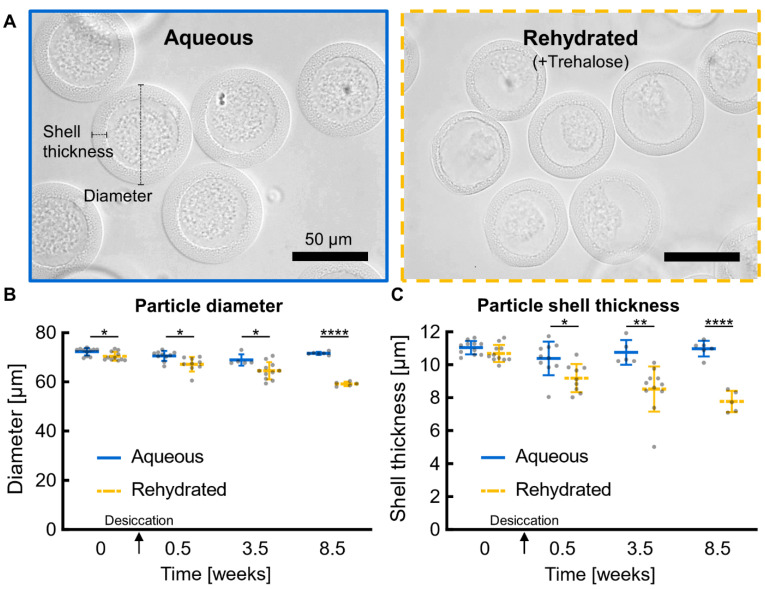
**Particle morphology after desiccation and rehydration.** Particle morphology was assessed by appearance, particle diameter, and particle shell thickness. (**A**) Rehydrated PicoShells (+Trehalose) stored for 3.5 weeks after desiccation look similar to fresh aqueous PicoShells (0 weeks). All scale bars are 50 μm. (**B**) Rehydrated particles exhibit slightly smaller diameters and (**C**) shell thicknesses compared to aqueous particles stored in PBS. Note that the 0-week timepoint for “Rehydrated” particles represents +Trehalose particles that were phase transferred into aqueous phase instead of being desiccated; the 0.5-week timepoint is the first post-desiccation timepoint. Raw data are overlaid on a line and error bars showing the mean and one s.d. (*n* = 5–11 particles). * *p* ≤ 0.05, ** *p* ≤ 0.01, **** *p* ≤ 0.0001.

**Figure 3 life-12-01168-f003:**
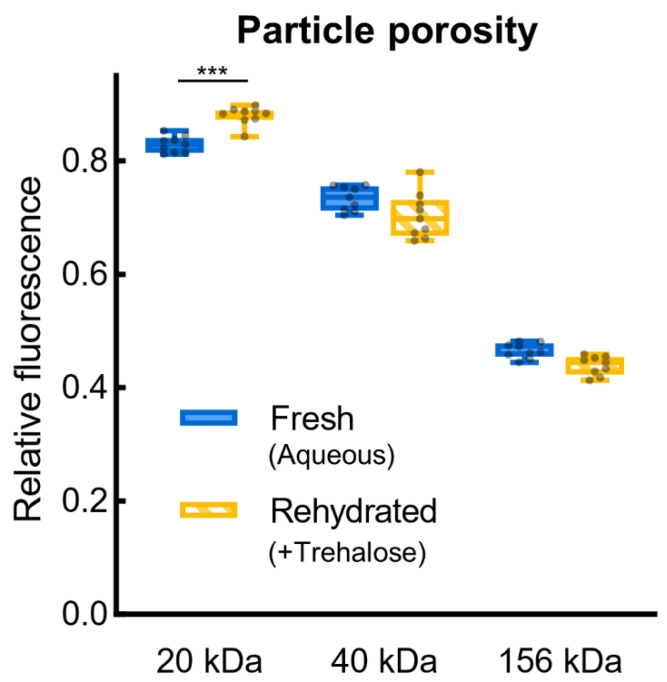
**Hydrogel shell porosity.** Fresh aqueous PicoShells were assayed immediately after particle fabrication and phase transfer (0 weeks), while rehydrated PicoShells (+Trehalose) were measured after desiccation and storage for 3.5 weeks. Particles were incubated for 20 min with 3 differently sized FITC-dextran molecules—20 kDa, 40 kDa, and 156 kDa FITC-dextran. Comparisons between molecular weights (not shown) are all significant (*p* < 0.0001). Data are plotted as a box and whiskers plot with the box extending to the 25th and 75th percentiles, the line in the middle of the box showing the median, the whiskers extending to the minimum and maximum value, and the raw data overlaid (*n* = 9 particles). *** *p* ≤ 0.001.

**Figure 4 life-12-01168-f004:**
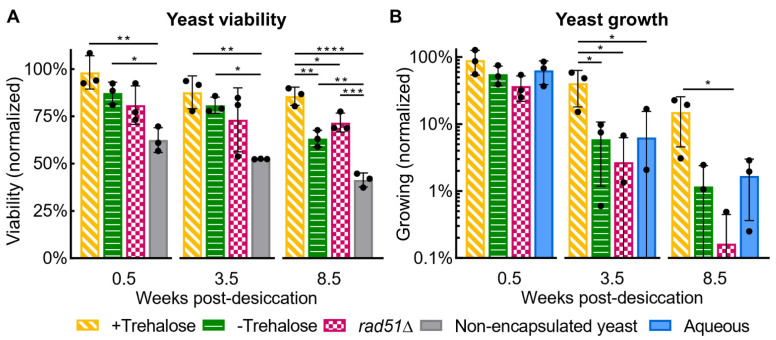
**Yeast health**. Yeast health was quantified via viability and growth potential at three post-desiccation timepoints. (**A**) The viability of each replicate is normalized to the viability of the same biological replicate that was phase transferred into PBS shortly after particle fabrication and assayed immediately (0 weeks). (**B**) Growth is normalized in the same way as viability and shows the portion of the population that grew to fill or swell the PicoShell after incubation for 16 h. Where fewer than three datapoints are shown, the unmarked datapoints had no growing yeast in any particles. Data are plotted as bars showing the mean and one s.d. as well as biological replicate datapoints (*n* = 3 wells). Significance markers are shown only within each timepoint. * *p* ≤ 0.05, ** *p* ≤ 0.01, *** *p* ≤ 0.001, **** *p* ≤ 0.0001.

**Figure 5 life-12-01168-f005:**
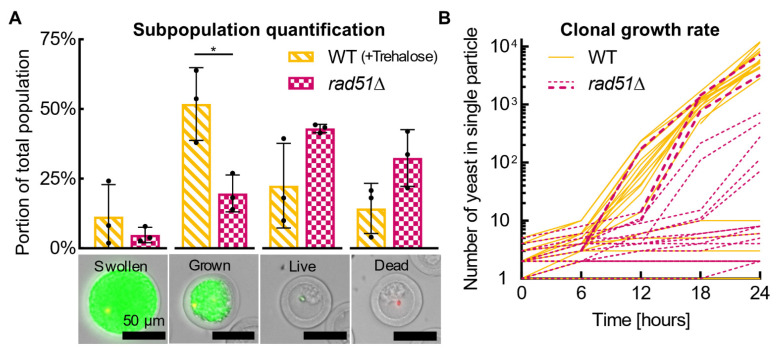
**Single-cell colony growth analysis**. Wild type (+Trehalose) and *rad51*Δ yeast in particles were rehydrated and compared 0.5 weeks after desiccation via subpopulation and clonal growth rate analyses. (**A**) Each population was divided into 4 subpopulations based on images of post-incubation, live-dead stained particles—“swollen” (yeast stretch particle), “grown” (yeast fill particle), “live” (yeast are alive, but not grown or swollen), and “dead”. The subpopulations add to 100%. Data are plotted as a bar plot showing the mean and one s.d. as well as biological replicate datapoints (*n* = 3 wells). Each biological replicate population comrpises 150–420 individual particles with yeast. All scale bars are 50 μm. (**B**) Growth trajectory of individual WT (+Trehalose) and *rad51*Δ yeast cells was tracked within 20 particles each. Some particles appeared or disappeared from view part-way through; these were included. Thicker dashed *rad51*Δ lines are only for easier viewing. Data are plotted as a line plot showing individual data points. * *p* ≤ 0.05.

## Data Availability

Data are available upon request.

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
