# Peer review of "Enabling Clonal Analyses of Yeast in Outer Space by Encapsulation and Desiccation in Hollow Microparticles"

_life, 2022, doi:10.3390/life12081168_

Round 1
Reviewer 1 Report
The authors developed a method to desiccate PicoShells with encapsulated single S. 112 cerevisiae cells that maintain particle morphology and yeast health after storage and rehydration. The quantitative analysis, particle diameters, and shell thicknesses of particle morphology was characterized by Imaging Microscope. The manuscript is well organized. The presented paper is interesting, but the following corrections should be made before publishing.
Below are my concerns and suggestions to improve the manuscript,
The abstract needs to be highly quantitative.
Please provide more details about section 2.2. PicoShell fabrication.
Please, explain the thermal stability and chemical stability of the particles.
Author Response
Thank you! We appreciate your constructive comments and suggestions. See our responses below.
1. The abstract needs to be highly quantitative.
We appreciate this comment and have added more quantitative metrics to our abstract, drawn from the Results section. See the revised abstract for details (lines 26-33).
2. Please provide more details about section 2.2. PicoShell fabrication.
We have added additional details on PicoShell fabrication to section 2.2 PicoShell fabrication and yeast encapsulation, including a reference to an influential protocol paper on microfluidic device fabrication, description of the hydrophobic surface treatment for microfluidic devices, and details on reagent pumping during PicoShell fabrication (lines 140-167).
3. Please, explain the thermal stability and chemical stability of the particles.
We thank the reviewer for identifying this gap and have provided information in 4.3 Potential future applications about intentional degradation mechanisms with PicoShells (lines 594-605), which can be used to release selected cells after sorting. We have also elaborated on thermal and chemical stability (lines 599-605). In short, PicoShells should be stable in a variety of solvents and at high temperatures, at least for short durations, though there is no comprehensive characterization yet as PicoShells are still a nascent technology.
Sincerely,
Sergio R. Santa Maria
Reviewer 2 Report
In this article, Ng and colleagues have investigated the use of PicoShells for encapsulation, storage, and desiccation of yeast samples in outer space. PicoShells have been previously developed by Di Carlo’s group and found many functional applications. In this study, the authors investigate the potential of PicoShells for desiccation and rehydration. The article looks nice and is well-structured. They are some specific comments that require attention
1. In the introduction section, line 59, the authors mentioned that “There are numerous single-cell analysis methods, each with their own applications and limitations. Two well-established single-cell analysis methods are flow cytometry and fluorescence activated cell sorting (FACS), which can characterize and sort single cells by cell size and/or presence of a fluorescent marker” this seems not to be well-written. FACS is not a single-cell analysis method; rather, it is a cell sorting approach. So it is better not to categorize FACS as a single cell analysis method.
2. In Fig. 1, it seems the authors used the same picture for B2 and B3. It is recommended to substitute the B3 with another one.
3. What is the empty size of PicoShells, and compared to the yeast, how many cells occupy each of the PicoShells? In each sampling, how many single cells, doublets, triplets, and more do they have? Is it essential to have the yeast as a single cell or not?
4. It is recommended for authors to provide an SEM image before and after the process of desiccation to ensure PicoShells preserve their functionality.
5. For shell porosity measurement, the authors investigate the molecular diffusion of FITC-dextran of different molecular weights, including 20, 40, and 156 kDa. However, if PicoShells have defects, like cracks, then the diffusion also happens. So I believe the proper investigation of the PicoShells is required to ensure all subsequent experiments are valid.
6. Are the PicoShell particles unaffected by doing the pipetting? Especially in the case of 8.5 weeks, where vigorous pipetting is required to ensure particles are separated. How does vigorous pipetting affect cell viability?
7. In Figure 4A, how do authors normalize the viability? Actually, what does normalized viability mean?
Author Response
Thank you for the insightful comments and suggestions. See our responses below.
1. In the introduction section, line 59, the authors mentioned that “There are numerous single-cell analysis methods, each with their own applications and limitations. Two well-established single-cell analysis methods are flow cytometry and fluorescence activated cell sorting (FACS), which can characterize and sort single cells by cell size and/or presence of a fluorescent marker” this seems not to be well-written. FACS is not a single-cell analysis method; rather, it is a cell sorting approach. So it is better not to categorize FACS as a single cell analysis method.
This is an astute distinction. We have changed the language of this section (lines 63-66) so that FACS is not described as a single-cell analysis method, but rather as a single-cell tool, which we believe to be sufficiently broad, as tools can have multiple uses, while a method gives the connotation of leading to a single goal.
2. In Fig. 1, it seems the authors used the same picture for B2 and B3. It is recommended to substitute the B3 with another one.
We thank the reviewer for pointing this out. In fact, these are different images, though they are very similar due to the lack of morphological change immediately after desiccation versus after an additional 8.5 weeks of storage. Please note the minor differences, such as the dots beneath the scale bar. We have added additional language in the caption to make it more explicitly clear that these are not the same image (lines 336-338).
3. What is the empty size of PicoShells, and compared to the yeast, how many cells occupy each of the PicoShells? In each sampling, how many single cells, doublets, triplets, and more do they have? Is it essential to have the yeast as a single cell or not?
These are all great questions, and we have added additional description to section 2.2 PicoShell fabrication and yeast encapsulation (lines 155-167). Empty PicoShells are ~70 μm, while yeast are 2-5 μm. Typically, a single cell occupies each PicoShell since the yeast is at a diluted concentration during particle fabrication. At most, a particle will have a few cells, often when cells have budded from each other or are otherwise stuck together. It can be problematic for single-cell analysis results if there is more than one cell in a PicoShell that did not directly bud from each other, but there are potential methods to remove these PicoShells from the pool, as discussed in 4.3 Potential future applications (lines 581-593). For consistency, we removed specific percentages of particles with single cells versus empty PicoShells from 4.3 Potential future applications, as these percentages were estimates from a previous publication which used a different cell concentration during encapsulation.
4. It is recommended for authors to provide an SEM image before and after the process of desiccation to ensure PicoShells preserve their functionality.
We thank the reviewer for the recommendation. SEM would provide interesting insight into the physical structure of the hydrogel mesh before and after desiccation, but we are unable to acquire SEM images at this point. In any case, SEM would not prove that PicoShells preserve functionality as well as testing directly for the desired functionality. Instead, we show that rehydrated PicoShells retain functionality via our FITC-dextran diffusion experiment (Figure 3), which shows that rehydrated PicoShells are physically intact and that the effective porosity to reagents in solution is not drastically changed from non-desiccated PicoShells.
5. For shell porosity measurement, the authors investigate the molecular diffusion of FITC-dextran of different molecular weights, including 20, 40, and 156 kDa. However, if PicoShells have defects, like cracks, then the diffusion also happens. So I believe the proper investigation of the PicoShells is required to ensure all subsequent experiments are valid.
In rare cases where a particle has ruptured, the particle immediately fills with the fluorescent solution in a manner clearly different than intact particles. We added discussion to 3.2.1 Particle integrity (lines 411-413) and added a panel to Figure S4 (Figure S4d) including a ripped particle alongside other particles that look very different from the ripped particle, indicating that most particles are not ripped. Even if desiccated particles that appeared intact do have small rips, we did not observe yeast escaping the particle and the diffusion of molecules across the particle shell is similar to non-desiccated particles (see Figure 3), so these particles satisfy the critical functions.
6. Are the PicoShell particles unaffected by doing the pipetting? Especially in the case of 8.5 weeks, where vigorous pipetting is required to ensure particles are separated. How does vigorous pipetting affect cell viability?
Individual PicoShells broken away from large clumps after vigorous pipetting retain good circularity, shell morphology, and porosity. It is unclear if the pipetting affects cell viability, since increased storage duration corresponds with increasingly vigorous pipetting, and both could decrease cell viability. We have made notes about these topics in sections 3.2.1 Particle integrity (lines 431-433) and 4.4 Limitations (lines 640-644).
7. In Figure 4A, how do authors normalize the viability? Actually, what does normalized viability mean?
We have added additional description of the normalization process in section 2.7 Viability and Growth which clarifies our normalization calculation (lines 241-249). Normalized viability and growth rate make it easier to compare the change in viability and growth rate between the tested conditions. For example, at the pre-desiccation timepoint for yeast growth (% of yeast population that divides), the rad51? condition had 67% of the population able to grow, while the +Trehalose (wild type yeast) had 71% and the -Trehalose (wild type) had 77%. The rad51? cells are generally expected to be less healthy than WT cells, but we are more interested in how desiccation affects each condition rather than the baseline differences between conditions. That is why we normalize, though we do show the non-normalized data in Figure S7 so that readers can also make absolute comparisons if desired.
Sincerely,
Sergio R. Santa Maria
Reviewer 3 Report
Studying microbes at the single-cell level can help uncover cellular heterogeneity and screen for highly performing cells. In this manuscript, Ng et al., developed a method, called PicoShells, to encapsulate microbes in nanoliter-scale compartment using microfluidic technology. The reported PicoShells enabled analysis of single cell derived colonies and the accompanying desiccation method preserved high viability of yeast. The results are convincing and clearly present. The manuscript is well written. The Reviewer finds high enthusiasm in this study and strongly suggest accepting this manuscript.
Author Response
Thank you! We appreciate your positive comments about our manuscript.
Sincerely,
Sergio R. Santa Maria
Round 2
Reviewer 1 Report
Authors have carefully checked and modified this manuscript. Now it can be accepted for publication in this journal without further revision.
Reviewer 2 Report
The authors have addressed all my comments